# Prevalence and Characterization of the Antimicrobial Resistance and Virulence Profiles of *Staphylococcus aureus* in Ready-to-Eat (Meat, Chicken, and Tuna) Pizzas in Mansoura City, Egypt

**DOI:** 10.3390/antibiotics14080817

**Published:** 2025-08-10

**Authors:** Sara Amgad Elsalkh, Amira Ibrahim Zakaria, Samir Mohammed Abd-Elghany, Kálmán Imre, Adriana Morar, Khalid Ibrahim Sallam

**Affiliations:** 1Department of Food Hygiene, Safety, and Technology, Faculty of Veterinary Medicine, Mansoura University, Mansoura 35516, Egypt; saraamgad2004@mans.edu.eg (S.A.E.); amirazakaria@mans.edu.eg (A.I.Z.); drsamir@mans.edu.eg (S.M.A.-E.); 2Department of Animal Production and Veterinary Public Health, Faculty of Veterinary Medicine, University of Life Sciences “King Mihai I” from Timisoara, 300645 Timisoara, Romania; adrianamo2001@yahoo.com

**Keywords:** pizza, MRSA, VRSA, enterotoxins, multidrug-resistant *S. aureus*

## Abstract

**Introduction:** *Staphylococcus aureus* is a high-priority foodborne pathogen contributing to several food poisoning outbreaks. Methicillin- and vancomycin-resistant *S. aureus* (MRSA and VRSA), pose significant public health concerns due to their potential for serious illness, antibiotic resistance, and transmission within both healthcare and community settings. These bacteria can cause numerous infections, ranging from skin and soft tissue infections to life-threatening conditions like bloodstream infections, pneumonia, and endocarditis. Although several publications are concerned with *Staphylococcus aureus* contamination in ready-to-eat (RTE) food products, little published data is available about its prevalence in pizza, which is widely distributed and consumed worldwide. **Methods:** The current study is intended to determine the prevalence, virulence genes, and antimicrobial resistance profiles of *S. aureus* in three hundred ready-to-eat pizza samples (100 each of meat, chicken, and canned tuna pizzas) collected from different restaurants in Mansoura City, Egypt. The typical colonies on Baird–Parker selective agar supplemented with egg yolk tellurite emulsion were counted and further confirmed based on Gram staining, coagulase testing, catalase testing, carbohydrate fermentation, and thermostable nuclease production. The genomic DNA of the confirmed coagulase-positive isolates was prepared and subjected to PCR analyses for detecting the *nuc* gene, *mec*A (methicillin resistance gene), and vancomycin resistance gene (*vanA*), as well as six selected *S. aureus* virulence genes: *sea*, *seb*, *sec*, *sed*, *hla*, and *tsst*. The antimicrobial resistance profile of the *S. aureus* isolates was determined against 16 antimicrobial agents belonging to six classes using the agar disc diffusion method according to the Clinical and Laboratory Standards Institute guidelines (CLSI), except for oxacillin and vancomycin, which were assessed using the MIC test. **Results:** The results revealed that 56% (56/100), 56% (56/100), and 40% (40/100) of chicken, meat, and canned tuna pizzas were positive for *S. aureus*, with an overall prevalence of 50.7% (152/300). All 560 isolates (100%) were verified as *S. aureus* based on molecular confirmation of the *nuc* gene. Interestingly, 48.6% (272/560) and 8.6% (48/560) of the isolates tested were identified as methicillin- and vancomycin-resistant *S. aureus* (MRSA and VRSA) through detection of *mecA* and *vanA* genes, respectively. Among the *S. aureus* isolates tested, the *hla* gene was detected in 87.1% (488/560), while the enterotoxin genes *sea*, *seb*, *sec*, and *sed* were identified in 50% (280/560), 78.6% (440/560), 9.8% (55/560), and 24.5% (137/560) of isolates, respectively. All recovered isolates (*n* = 560) were classified as multidrug-resistant and were resistant to penicillin, oxacillin, and ampicillin. Moreover, 77% (431/560), 24% (134/560), 8% (45/560), and 8.6% (48/560) of isolates were resistant to cefotaxime, ciprofloxacin, azithromycin, and vancomycin, respectively. **Conclusions:** The current study emphasizes that ready-to-eat pizza is highly contaminated with multidrug-resistant *S. aureus*, highlighting the urgent need for rationalizing antibiotic use in both veterinary and human medicine to prevent the transmission of resistant bacteria through the food chain. Additionally, strict adherence to good hygienic practices throughout all stages of the food chain is essential to minimize overall contamination and enhance food safety.

## 1. Introduction

The demand for ready-to-eat meals appears to be increasing globally in recent years due to limited time for meal preparation and consumption, along with the increased hours that people spend outside their homes, whether women, men, or even children and youth. Pizza is one of the most popular ready-to-eat meals due to its unique flavor, delicious taste, and affordable price, along with its nutritional value, as it provides consumers with the three main macronutrients: protein, carbohydrates, and fats.

Pizza is a fairly simple dish that consists of three basic elements: crust, sauce, and topping. The crust is similar to bread dough; the sauce is a paste of tomatoes, oil, water, and seasonings; and the topping of pizza has endless options, including cheese, mushrooms, olives, chicken, meat or meat products like pepperoni, and seafood such as shrimp, crab, and tuna. Pizza is cooked in a hot oven at 180 °C until the crust is baked and the cheese is melted.

Ready-to-eat food is highly susceptible to contamination, starting from the quality of raw materials, air, dust, unclean utensils, poor personal hygiene of workers, insufficient cooking temperature, and post-processing contamination. Fast food consumption has been associated with slower economic growth and serious health problems globally due to its high bacterial contamination levels during processing [1]. Additionally, the high number of workers involved in food processing and the high rate of pathogenic bacteria contaminating the food, including *Staphylococcus aureus*, especially under poor personal hygienic conditions and unsafe temperature control, are major concerns [1].

*S. aureus* is a Gram-positive bacterium and facultative anaerobe found as a commensal opportunistic flora in the upper respiratory tract, mucous membrane, and on the skin. *S. aureus* is differentiated from other staphylococci by the coagulase test, as all *S. aureus* strains are coagulase-positive. Although most researchers have reported that animal-derived foods are often found to be contaminated with coagulase-positive *S. aureus*, recently, the pathogenicity of coagulase-negative staphylococci has been reported in many studies as a serious pathogen causing many fatal infections in both animals and humans [2].

*Staphylococcus aureus* is a high-priority foodborne pathogen and has been continuously studied by many researchers worldwide due to its pathogenicity, production of heat-stable enterotoxins, and rapid development of multidrug resistance. Special attention is required due to easy food contamination with *S. aureus* during food handling, processing, and packaging, as it is a common commensal of the skin and mucosal membranes of the nose and throat in 40% of humans, especially under poor hygienic conditions. It is the most reliable indicator of inefficient thermal treatment of any food, unhygienic circumstances, and slow cooling after the preparation of a meal. Nonetheless, the risk associated with staphylococci is not limited to their pathogenic properties since they show lipolytic and proteolytic activity in various temperature and salinity conditions, so they can cause both food spoilage and poisoning [3].

*S. aureus* is classified as the third most common cause of foodborne illness in many countries, including Brazil, Egypt, Taiwan, and Japan [4]. Additionally, it results in 241,000 foodborne illnesses per year in the United States [5]. *Staphylococcus* foodborne disease usually leads to significant fluid and electrolyte loss due to severe, abrupt vomiting and diarrhea in infected patients. Physical examination may reveal signs of dehydration and hypotension; however, in this situation, the disease is self-limiting within 24−48 h except in infants, the elderly, and immunocompromised people [6].

*S. aureus* has various virulence factors that attack their host, including cytotoxins that which are incorporated in adherence, colonization, ability to survive inside invasive tissues, exfoliative toxins A and B, Panton–Valentine leukocidin (PVL), hemolysins (alpha, beta, gamma, and delta), and toxic shock syndrome toxin-1 (TSST-1) [3,7]. Furthermore, the pathogenicity of this organism extends to the production of many enterotoxins in food, which are classified as the most dangerous toxins worldwide, as they resist both heat and digestive enzymes, retaining their activity [8] with resultant staphylococcal foodborne intoxication. The most common types of these toxins are SEA, SEB, SEC, SED, and SEE, which contribute to about 95% of *Staphylococcus* intoxication outbreaks from dairy products [9,10].

The misuse and abuse of antibiotics in humans and animals have led to the development of antimicrobial resistance genes in bacteria, with a particular focus on *S. aureus*, which has acquired multiple resistance genes to the most clinically effective antimicrobials, resulting in treatment failures in many patients. This resistance is created through the continuous mutation of chromosomal genes or the acquisition of resistance determinants passed horizontally [11].

The consumption of food contaminated with *S. aureus* bacteria is considered a source of potential risk for consumers. Methicillin-resistant *S. aureus* (MRSA) is an emergent public health problem. In normal cases, a patient with *S. aureus* infection can be treated with ß-lactam antibiotics, mainly oxacillin and/or cefoxitin. ß-lactam antibiotics can bind with penicillin-binding proteins (PBPs), which are essential for the synthesis of the bacterial cell wall as well as the formation of the peptidoglycan crosslink that results in the lysis of the bacterial cell. In the case of MRSA, the *mec*A gene (carried by a mobile genetic element called the staphylococcal cassette chromosome) is expressed by the production of PBP2a (modified PBP) with a markedly reduced affinity for β-lactam antibiotics, and consequently, MRSA strains can overcome the suppression of cell wall production [12].

Although much research concerning *S. aureus* in ready-to-eat (RTE) meat products is published, data is scarce on *S. aureus* contamination in pizzas. The current study was therefore carried out to highlight the interest in the prevalence of MRSA in retail chicken, meat, and tuna pizzas marketed in Egypt and to characterize the virulence genes (*sea*, *seb*, *sec*, *sed*, and *tsst*-1) and the antimicrobial resistance profiles of the strains isolated during this survey.

## 2. Results

### 2.1. Prevalence and Count of Staphylococcus aureus in Pizza Samples

The increasing consumption rate of RTE pizza worldwide encourages scientists to focus on assessing its microbiological quality, especially for *S. aureus*, which is considered one of the most threatening foodborne pathogens. For this reason, 300 samples of meat, chicken, and tuna pizza were examined to determine the degree of their contamination. Of the three pizza types tested (*n* = 300), *S. aureus* was identified in 50.7% (152/300) of samples, including 56% (56/100) of meat pizza, 56% (56/100) of chicken pizza, and 40% (40/100) of canned tuna pizza (Figure 1A), with mean counts of 4.61, 6.4, and 4.64 log_10_ CFU/g, respectively (Figure 1B).

Interestingly, 48% (48/100), 52% (52/100), and 40% (40/100) of the examined meat, chicken, and canned tuna pizza samples, respectively, exceeded the maximum permissible limit of 10^3^ CFU/g issued by Egypt’s National Food Safety Authority (NFSA) [13] and were considered unacceptable for human consumption (Table 1).

### 2.2. Molecular Characterization of S. aureus Isolates Recovered from Examined Pizza Samples

#### 2.2.1. Identification of Methicillin- and Vancomycin-Resistant *S. aureus* (MRSA and VRSA) Among *S. aureus* Isolates

All of the 560 *S. aureus* isolates (264 from chicken, 160 from meat, and 136 from tuna pizza) recovered from the *S. aureus*-positive samples (152/300; 50.7%) were molecularly confirmed based on the *S. aureus* marker gene, the thermonuclease (*nuc*) gene, which was detected at the expected molecular size of 278 bp DNA fragment (Figure 2A).

In the present study, 35% (35/100), 43% (43/100), and 24% (24/100) of the examined meat, chicken, and canned tuna pizza samples, with an overall prevalence rate of 34% (102/300), were positive for MRSA. Concerning the isolates, 45% (72/160), 42.4% (112/264), and 64.7% (88/136) of *S. aureus* isolates recovered from RTE meat, chicken, and tuna pizza, respectively, with an overall incidence of 48.6% (272/560), were confirmed as MRSA, as they were positive for the *mecA* gene, as determined by PCR at the expected molecular size of 1200 bp (Figure 2B). On the other hand, 15% (15/100), 6% (6/100), and 4% (4/100) of the examined meat, chicken, and canned tuna pizza samples, with an overall prevalence rate of 8.3% (25/300), were positive for VRSA. At the isolate level, 20% (32/160), 3.03% (8/264), and 5.9% (8/136) of *S. aureus* isolates recovered from RTE meat, chicken, and tuna pizza, respectively, with an overall prevalence rate of 8.6% (48/560), were confirmed as VRSA, as they were positive for the *vanA* gene, as determined by PCR at the expected molecular size of 235 bp (Figure 2C).

The prevalence and distribution of methicillin-resistant *S. aureus* (MRSA) and vancomycin-resistant *S. aureus* (VRSA) among the nuclease (*nuc*)-positive *S. aureus* isolates (*n* = 560) recovered from the examined samples of meat, chicken, and canned tuna pizza is demonstrated based on the sample level (Figure 3A) and based on the isolate level (Figure 3B).

#### 2.2.2. Screening of Different Virulence Genes Among *S. aureus* Isolates Recovered from Meat, Chicken, and Canned Tuna Pizza

The 560 *S. aureus* isolates (160 from meat, 264 from chicken, and 136 from tuna pizza) recovered in the present study were molecularly screened by multiplex PCR for different virulence factors, including SE genes such as *sea*, *seb*, *sec*, and *sed*, which were identified at the expected molecular sizes of 500, 164, 451, and 278 bp, respectively (Figure 4A).

Twelve existence and coexistence patterns of the four different enterotoxins investigated in enterotoxigenic *S. aureus* isolates were retrieved in the present study. A quadruple pattern was present in seven *S. aureus* isolates, showing the coexistence of the four enterotoxin genes *sea*, *seb*, *sec,* and *sed*, as demonstrated in Lane 1 (Figure 4A). Four triple patterns (Lanes 2–5; Figure 4A) revealed the coexistence of three enterotoxin genes, namely, *sea*, *seb*, and *sec*; *sea*, *seb*, and *sed*; *sea*, *sec,* and *sed*; and *seb*, *sec,* and *sed* in 13, 19, 9, and 13 *S. aureus* isolates, respectively. Four duplicate patterns (Lanes 6–9; Figure 4A) showed the coexistence of two enterotoxin genes, namely, *sea* and *seb*; *sea* and *sec*; *sea* and *sed*; *seb* and *sed* in 145, 13, 27, and 41 *S. aureus* isolates, respectively. The last three patterns (Lanes 10–12; Figure 4A) revealed the existence of a single enterotoxin gene alone, namely *sea*, *seb*, and *sed*, in 47, 202, and 21 *S. aureus* isolates, respectively.

The prevalence and distribution of enterotoxin genes (*sea*, *seb*, *sec*, and *sed*) and the α-hemolysin gene (*hla*) among *S. aureus* isolates (*n* = 560) recovered from the examined samples of meat, chicken, and canned tuna pizza are shown in Figure 5.

*Staphylococcus* enterotoxin B (*seb*) was the most frequently detected enterotoxin gene in all isolates from different pizza samples tested, detected in 72.5% (116/160), 79.5% (218/264), and 82.6% (106/136) of *S. aureus* isolates recovered from meat, chicken, and canned tuna pizza, respectively with an overall incidence of 78.6% (440/560). The *sea* gene was the second most prevalent *S. aureus* enterotoxin, detected at incidences of 52.5% (84/160), 46.2% (122/264), and 54.4% (74/136) in *S. aureus* isolates recovered from meat, chicken, and tuna pizza, respectively, with an overall incidence of 50% (280/560), indicating the risk of food poisoning caused by *S. aureus* enterotoxin in Egypt.

Moreover, enterotoxin type C (*sec*) was detected in 6.3% (10/160), 6.8% (18/264), and 19.9% (27/136) of *S. aureus* isolates recovered from meat, chicken, and tuna pizza, respectively, with an overall incidence of 9.8% (55/560). Enterotoxin *D* (*sed*), however, was detected in 11.3% (18/160), 25.8% (68/264), and 37.5% (51/136) of *S. aureus* isolates recovered from meat, chicken, and tuna pizza samples, respectively, with an overall incidence of 24.5% (137/560).

The 560 *S. aureus* isolates were further screened by PCR for the existence of the *hla* gene, which was detected at the expected molecular size of 704 bp (Figure 4B). The prevalence and distribution of the α-hemolysin gene (*hla*) among *S. aureus* isolates (*n* = 560) recovered from the examined samples of meat, chicken, and canned tuna pizza in the present investigation are shown in Figure 5. The *hla* gene was detected in 83.1% (133/160), 86.7% (229/264), and 92.6% (126/136) of meat, chicken, and canned tuna pizza, with an overall detection incidence of 87.1% (488/560) (Figure 5). On the other hand, testing of the 560 *S. aureus* isolates recovered in the present study for the *tsst* gene indicated that all (100%) of the isolates tested were *tsst* gene negative.

### 2.3. Antimicrobial Resistance Profile of S. aureus Isolated from Different Types of Pizza

In the present study, all (100%) of the 560 *S. aureus* isolates tested were resistant to ampicillin, oxacillin, and penicillin (Table 2), which are the primary antimicrobial drugs administered for the treatment of Gram-positive bacterial infection. This study also demonstrated high resistance rates of 87% (487/560), 77% (431/560), 76% (426/560), and 24% (134/560) for *S. aureus* isolates toward cefuroxime, cefotaxime, kanamycin, and ciprofloxacin, respectively (Table 2), along with low resistance rates of 14% (78/560), 8.6% (48/560), 8% (45/560), 7% (40/560), and 7% (40/560) for *S. aureus* isolates against tetracycline, vancomycin, azithromycin, sulfamethoxazole-trimethoprim, and clindamycin, respectively (Table 2). Interestingly, none of the tested isolates were resistant to linezolid or rifampin (Table 2).

In the present study, 87% (487/560) and 77% (431/560) of *S. aureus* isolates examined were resistant to cefuroxime and cefotaxime, respectively (Table 2). Testing against aminoglycosides, however, revealed that 76% (426/560) and 4% (12/560) of *S. aureus* isolates were resistant to kanamycin and gentamicin, respectively (Table 2).

Additionally, 24% (134/560) of *S. aureus* isolates tested were resistant to ciprofloxacin, while only 14% (78/560) of recovered *S. aureus* isolates were resistant to tetracycline. On the other hand, *S. aureus* isolates had a resistance rate of 8% (45/560) to azithromycin and 7% (40/560) to sulfamethoxazole-trimethoprim. Likewise, a resistance rate of 7% (40/560) was detected in *S. aureus* isolates against clindamycin, and 8.6% (48/560) of *S. aureus* isolates were phenotypically resistant to vancomycin, all of which were verified by PCR to harbor the *vanA* gene, the most common genetic element responsible for resistance against vancomycin. Interestingly, linezolid and rifampin encountered a complete absence of resistance in our survey, since all of the 560 (100%) *S. aureus* isolates were susceptible to both antibiotics (Table 2).

Classification of *S. aureus* isolates (*n* = 560) based on their multiple antibiotic resistance (MAR) index and antimicrobial resistance profiles against the 16 antimicrobial agents tested is shown in Table 3. Interestingly, all *S. aureus* strains tested in this study were categorized as multidrug-resistant (MDR) pathogens with an MAR index ranging from 0.2 to 0.7 based on their resistance profiles against the 16 antimicrobials tested (Table 3).

Altogether, the revealed overall antimicrobial resistance pattern of enterotoxigenic *S. aureus* strains can be considered disquieting in the present survey, widely agreeing with many results of other studies conducted in other regions worldwide with various hygienic measures. Therefore, it is crucial to establish monitoring systems that impose a rational usage of antibiotics in veterinary medicine to protect public health from the spread of multidrug-resistant bacteria to humans via animal-origin foodstuffs. Also, this study is a significant call for health authorities to enact laws on food outlets, restaurants, and street vendors to apply strict hygienic measures to protect consumers from serious pathogens.

## 3. Discussion

### 3.1. Prevalence of Staphylococcus aureus in Pizza Samples in Comparison with RTE Meat Products in Other Studies

The high prevalence of *S. aureus* in the examined pizza samples in the present study indicated that prepared pizza may be subjected to post-cooking contamination from the pizza handlers, as it may be left aside for some time at room temperature before being packaged and delivered to consumers. Moreover, the cultural, educational, and sanitary background of food handlers have a critical effect on the degree of food contamination in food outlets. The prevalence rates and counts of *S. aureus* in RTE meat products vary from one study to another worldwide. Similar *S. aureus* prevalence rates of 50.8% (62/120) were detected in RTE meat products sampled from Benha, Egypt, with a mean count of 3.2 log_10_ CFU/g [14]. Also in Turkey, 56.3% (45/80) of RTE meat sandwiches were positive for *S. aureus* with a mean count of 3.3 log_10_ CFU/g [15]. Moreover, *S. aureus* was detected in 55.6% of cooked ground meat tested in Tehran, Iran [16]. In contrast, *S. aureus* was isolated at a lower prevalence rate of 40% from RTE meat products tested in Korea [17], while much lower prevalence rates of 20% (5/25) and 8% (2/25) were detected for *S. aureus* in beef and chicken burger sandwiches examined in Brazil, respectively [18]. Likewise, in China, only 25% (3/12) and 11.8% (54/456) of RTE beef and poultry products were contaminated with *S. aureus* [19]. Moreover, *S. aureus* could not be detected in grilled chicken sandwiches examined in Tehran, Iran [16]. On the contrary, a high prevalence rate of 83.1% (187/225) with a mean count of 4.0 × 10^3^ log_10_ CFU/g was determined for *S. aureus* in RTE meat sandwiches tested in Mansoura, Egypt [20]. Additionally, 70.3% of RTE meat product samples collected in Alexandria, Egypt, were positive for *S. aureus* with a mean count of 4.98 log_10_ CFU/g [21].

Canned tuna pizza in the present study showed a relatively lower prevalence rate of *S. aureus* at 40% (40/100). A prevalence rate of 44% was detected for *S. aureus* in imported canned tuna samples in Egypt [22]. The existence of such pathogens in canned products was explained by the contaminating manual handling of tuna fish before canning or a breakdown in thermal processing. In this context, Wu and Su [23] demonstrated that *S. aureus* could not be inhibited by freezing for up to 4 weeks before canning, and hence it could grow rapidly during thawing, reaching 3 log_10_ CFU/g. In contrast to our findings, Jang et al. [24] revealed that only 1.1% of canned fish sandwiches sold in South Korea were positive for *S. aureus*. The differences in the prevalence rates among the various studies from different countries may be attributed to variations in the contamination level of the raw ingredients, hygienic conditions, the addition of seasonings and sauces, the efficiency of cooking, and packaging, which all reflect the discrepancy of *S. aureus* prevalence in different investigations [25]. The high prevalence rate of contamination in the current study raises alarm regarding the food preparation process and its role in introducing one of the commensal opportunistic bacteria naturally found in the respiratory tract of humans in the food chain, which constitutes a public health concern that needs further studies.

### 3.2. Prevalence of Methicillin- and Vancomycin-Resistant S. aureus (MRSA and VRSA) in Pizza Samples in Comparison with Their Prevalence Rates in Other Studies

The *nuc* gene is not the sole specific marker for *S. aureus* detection, as other gene markers such as the *coa* gene, encoding coagulase, and additional genetic targets can also be employed for accurate identification. Our findings are in agreement with a previous study conducted by Mahros et al. [20] in Egypt, who reported that all *S. aureus* isolates recovered from RTE meat product sandwiches were positive for the *nuc* gene.

The relatively high recovery percentages of MRSA and VRSA from the RTE samples tested constitute major public health implications. The resistance of MRSA and VRSA to standard antibiotics makes infections more difficult to cure, posing serious public health concerns. MRSA is more widespread, while VRSA is a more recent and concerning development, highlighting the growing threat of antibiotic resistance.

Many studies worldwide have reported discrepancies in MRSA incidences among different foods. In Egypt, Abd El-Razik et al. [26] detected *mecA* in 100% of *S. aureus* isolated from RTE beef burger sandwiches. Conversely, a lower prevalence rate of 25% was reported for MRSA (*mecA* gene-positive *S. aureus*) among *S. aureus* isolates recovered from chicken-based meals [27]. In China, Wu et al. [19] isolated MRSA from 8.1% (8/99) of surveyed RTE meat. In the USA, Wells and Juett [28] detected MRSA in 18.5% (10/54) of meat samples collected from different sites in Kentucky. On the contrary, Safarpoor Dehkordi et al. [29] recorded that all *S. aureus* isolated from RTE meat and chicken barbecue collected from the big hospitals in Iran were methicillin-resistant. The high prevalence of the *mecA* gene in *S. aureus* derived from food of animal-origin samples may point to the fact that the gene first originated from strains of *Staphylococcus fleurettii*, which is found only in animal species [30].

In the last two decades, the mortality incidence from MRSA has exceeded deaths from other serious diseases such as hepatitis, tuberculosis, and AIDS. Moreover, MRSA displayed multiple antimicrobial resistances, and the WHO listed it among the 12 most dangerous bacterial families to human and animal health [31]. The mechanism of methicillin resistance by *S. aureus* is expressed either by the production of a huge amount of beta-lactamases, modification in the normal sites of penicillin-binding proteins (PBPs), or the synthesis of acquired penicillin-binding protein 2a (PBP2a) [19]. The PBP protein is encoded by the *mecA* gene, which is located in mobile genetic elements and facilitates its transfer either horizontally or vertically through bacterial generations [32].

Vancomycin was the drug of choice for the treatment of MRSA infection for many years and is still the selective drug for treatment now. However, many researchers have reported that some strains of *S. aureus* have developed intermediate or complete resistance to glycopeptide antimicrobial drugs, which creates an alarming threat [33]. The vancomycin-resistant gene first appeared in *Enterococcus* species in Europe; this resistance is mediated by mobile genetic elements primarily found on plasmids, facilitating their dissemination to susceptible medical pathogens, especially *S. aureus* [34]. The *vanA* gene is the primary genetic cluster among 11 clusters related to vancomycin resistance in *Enterococcus* species; it is the main reservoir of this gene [35].

Consistent with our results, Ghanem et al. [27] detected *vanA* in 12.5% of *S. aureus* isolated from cooked chicken samples served in Egyptian hospitals. However, Saber et al. [35] isolated VRSA from 26.7% of surveyed RTE shawarma and beef burger sandwiches in Zagazig City, Egypt. In Iran, Afshari et al. [36] detected three vancomycin-resistant variant genes, *vanA*, *vanB*, and *vanC*, in *S. aureus* isolated from meat and chicken meals with prevalence rates of 1.8%, 1.1%, and 20%, respectively. Nowadays, resistance to vancomycin, which is classified as the last resort to treat MRSA infection, is a critical point of concern for global health organizations and requires an emergent solution to preserve the lives of many people.

Extensive misuse of antibiotics in food-producing animals, either for therapeutic or productive purposes, in the last decades has been implicated in the development of antimicrobial resistance genes entering the human food supply chain. The spread of antimicrobial-resistant bacteria is an emergent threat worldwide, as it favors the evolution of diseases and reduces therapeutic options. *S. aureus* has developed multiple resistance genes to overcome the most commonly used antibiotics in the treatment of infection and encourage its growth and survival; therefore, the World Health Organization lists *S. aureus* as a high-priority bacterium that requires the development of new antimicrobials [37,38].

### 3.3. Comparison of the Virulent Genes of S. aureus Isolates Recovered from Meat, Chicken, and Canned Tuna Pizza with Those Detected in Other Studies

Staphylococcal enterotoxins (SEs) are encoded by their enterotoxin genes, which are described as a superpyrogenic family of exotoxins that share some structural and functional characteristics. There are about 28 SEs that have been identified up to now [39]. The secretion of these toxins begins when counts of bacteria reach 5–6 log_10_ CFU/g in the implicated food, as confirmed by Le Loir et al. [40]. This production process is induced by several environmental stress factors, including temperature, pH, osmotic pressure, and salt concentration [41]. Enterotoxins are proteins encoded on various genetic mobile elements in *S*. *aureus as* plasmids, prophages, bacterial pathogenicity islands, genomic islands, or next to staphylococcal cassette chromosome (SCC) elements, which enable the continuous modification of the dangerous pathogen to cause diseases [42].

The *sea* and *seb* genes encode staphylococcal enterotoxins A and B, respectively, which are the most common and potent enterotoxins produced by *S. aureus* and are involved in staphylococcal food poisoning. Both toxins are major causes of food poisoning, resulting in symptoms like nausea, vomiting, diarrhea, and abdominal cramps [42]. While SEA is not usually associated with toxic shock syndrome (TSS), SEB can contribute to this severe and potentially life-threatening illness. Both SEA and SEB act as superantigens, generating a massive and uncontrolled immune response through the activation of large numbers of T cells, leading to inflammation and tissue damage [42].

The SE genes are associated with most staphylococcal food intoxication outbreaks and can be used as a biological weapon due to their heat and proteolytic stability [43]. SEA is the most common type implicated in 90% and 77.8% of food poisoning outbreaks in Korea and the USA, followed by SEB [42]. In China, Wu et al. [43] detected SE genes in 108 *S. aureus* isolates with incidences of *sec* 75.0% (79/108), *sea* 63.9%, (68/108), *seb* 50.9% (54/108), and *sed* 12.96% (14/108) recovered from 4300 food samples, while Mahros et al. [20] revealed that 61.1% of *S. aureus* strains isolated from RTE meat product sandwiches in Egypt were positive for SE genes. In contrast, only one vegetable meal with meat among 5241 RTE samples tested in Turkey was positive for SEs, with a very low incidence of 0.019% [44].

Seb enterotoxin is a critical one among relevant enterotoxins, as only 20 ng in contaminated food with preformed enterotoxin can result in food poisoning symptoms [45]. On the contrary, the dose of any enterotoxin needed to cause food poisoning is about 1 µg, and in an outbreak due to enterotoxin (SEA)-contaminated food, 0.5 ng/mL was enough to produce poisoning manifestations [46]. SE proteins have powerful heat stability that can overcome cooking temperature, and these toxins tolerate digestive acidity and digestive enzymes like pepsin and trypsin, so they can outlast the bacteria that produce them in contaminated foods [47].

The different incidences and distributions of SEs and other virulence genes in different surveys are normal, as humans are the primary source of these pathogens, confirming that the contamination rates can increase in response to excessive food manipulation and preparation practices, which differ according to educational and traditional concepts in various food-serving places distributed worldwide. Moreover, the storage temperature, the environment of the food itself (pH, NaCl concentration, protein content, and water content), and post-cooking contamination with air, dust, and water can affect the food contamination rate, especially with enterotoxins, which are more heat-resistant than the producing bacteria themselves.

The *hla* gene is one of the lethal exotoxins described as dermonecrotic and neurotoxic in a wide range of hosts and is responsible for host cell homeostasis, affecting RBCs, endothelial cells, and epithelial cells. Similar to our findings for the *hla* gene, a high prevalence rate of 95.5% was also detected for the *hla* gene in the examined meat product sandwiches in Egypt [20]. Similarly, the *hla* gene was detected in 98% of RTE food in Switzerland [47], while a slightly lower prevalence rate of 84.3% was determined in *S. aureus* strains isolated from RTE chicken pastries tested in Oman [48].

One of the unique manifestations of *Staphylococcus* intoxication is staphylococcal toxic shock syndrome. This syndrome is manifested by fever, rash, and multiple organ failure [49]. The *TSST-1* is responsible for toxic shock syndrome, emitting huge amounts of procytotoxins from host T cells and macrophages [7]. In the present study, all (100%) of the 560 *S. aureus* isolates we tested did not encode the *tsst* gene. This finding is in agreement with those of a previous study conducted by Mahros et al. [20] in Egypt, who could not detect the *tsst* gene in any *S. aureus* isolates recovered from RTE meat product sandwiches. Other studies reported different frequencies of the *tsst* gene; Özdemir and Keyvan [50] reported that 7% of *S. aureus* isolates examined harbored the *tsst* gene from meat samples in Ankara, while Beshiru et al. [51] detected it in 20.8% of *S. aureus* isolates tested from RTE shrimp in Nigeria, confirming that the isolates encoded the *tsst* allele in a genetic structure usually related to serious infection.

### 3.4. Antimicrobial Resistance Profile of S. aureus Recovered from Pizza in Comparison with the Profile Determined in S. aureus Isolated from RTE Meat Products in Other Studies

In the last decades, the spread of multidrug-resistant bacteria has been described as an emerging public health threat that reduces the efficacy of antimicrobial treatment and contributes to the death of more than 25,000 people annually in the European Union [52]. The absolute resistance of all 560 (100%) *S. aureus* isolates tested to ampicillin, oxacillin, and penicillin agrees with the findings of Elshebrawy et al. [53], who recorded that all *S. aureus* isolates from chicken carcasses, RTE chicken sandwiches, and buffalo milk samples marketed in Mansoura City, Egypt, show complete resistance to penicillin. Similar results were obtained by Sallam et al. [54] in Egypt, who recorded that 93.4% and 88.9% of *S. aureus* recovered from chicken carcasses were resistant to penicillin and ampicillin. Likewise, Yang et al. [55] in China revealed that 98% of *S. aureus* isolates from RTE food were resistant to ampicillin. Conversely, only 5.8% and 22.6% of *S. aureus* isolated from RTE meat product sandwiches previously tested in our laboratory were resistant to ampicillin and oxacillin, respectively [20].

Cephalosporins have been used in veterinary medicine worldwide for the treatment of clinical mastitis and respiratory diseases. The resistance rates of 87% (487/560) and 77% (431/560) of *S. aureus* isolates observed against cefuroxime and cefotaxime, respectively, were lower than those detected by Parisi et al. [56] in Italy, who revealed that 100% of *S. aureus* isolates were resistant to cefoxitin and cefotaxime.

Aminoglycosides are widely used in livestock husbandry as broad-spectrum antibiotics or growth factors, which explains the high resistance rate to this category in food of animal origin. The resistance rates of 76% (426/560) and 4% (12/560) for *S. aureus* isolates toward kanamycin and gentamicin, respectively, in the present study are consistent with a previous study performed in China by Wu et al. [19], who reported that 75.9% and 27.8% of *S. aureus* isolates were resistant to kanamycin and gentamicin, respectively. The low resistance rate to gentamicin can be explained by its limited use in the veterinary field compared to kanamycin, since it is poorly absorbable through the intestinal walls and may cause allergic reactions through parenteral application [57].

Fluoroquinolones are extensively used to prevent or treat infection with both Gram-positive and Gram-negative bacteria. The resistance rate of 24% (134/560) reported in our study by *S. aureus* isolates against ciprofloxacin seemed slightly lower than that of Tang et al. [58] in Denmark, who noted that 32% of *S. aureus* recovered from retailed meat samples were resistant to ciprofloxacin. Conversely, 100% of *S. aureus* isolated from different slaughterhouses and meat outlets in Pakistan [59] and 100% of *S. aureus* isolated from beef samples examined in Georgia, USA, were resistant to ciprofloxacin [60].

Amazingly, only 14% (78/560) of recovered *S. aureus* isolates were resistant to tetracycline, which is used in livestock to treat many bacterial infections. This finding is similar to that reported in a previous study in Poland by Chajęcka-Wierzchowska et al. [61], who revealed that 17.9% of *S. aureus* isolates recovered from RTE foods were resistant to tetracycline. On the contrary, 68.4% (197/288) of *S. aureus* isolates recovered from retail chicken products in Egypt were resistant to tetracycline [54]. Conversely, none of the MRSA isolates tested from clinical human and pork samples tested in the USA were resistant to tetracycline [60].

Macrolides are a class of antimicrobials that act against a wide range of bacterial infections in humans and animals. The resistance rate of 8% (45/560) recorded for *S. aureus* isolates to azithromycin (the second generation of macrolides) seemed to be lower than the resistance rate of 50% detected in *S. aureus* isolated from RTE hamburgers, chicken nuggets, and salami in Iran [62]. Conversely, 100% of *S. aureus* isolates (*n* = 227) isolated from RTE shrimp in Japan were resistant to azithromycin [63].

The trimethoprim and sulfamethoxazole combination acts to deprive bacteria of folic acid, essential for growth; hence, their synergetic effect acts as a potent antimicrobial drug in animal treatment. The estimated resistance rate of 7% (40/560) identified in *S. aureus* isolates toward sulfamethoxazole-trimethoprim is slightly lower than that in a previous study conducted in China by Lin et al. [64], where a resistance rate of 11.1% was recorded against sulfamethoxazole-trimethoprim by *S. aureus* isolated from RTE meat and dairy products. On the other hand, a higher resistance rate of 27.9% was determined for sulfamethoxazole-trimethoprim in MRSA isolated from RTE food samples examined in Nigeria [65].

Likewise, a resistance rate of 7% (40/560) was detected in the present study for *S. aureus* isolates against clindamycin, although Wang et al. [25] revealed that all (*n* = 23) MRSA isolated from retail food samples, including chicken, RTE meat, cold noodles, dried tofu, and powdered infant formula, sold in 12 cities’ supermarkets in China were resistant to clindamycin.

Vancomycin is a glycopeptide antimicrobial, which is the last resort for treating MRSA infection. Our results revealed that 8.6% (48/560) of *S. aureus* isolates were phenotypically resistant to vancomycin, and all were verified by PCR to harbor the *vanA* gene, the most common genetic element responsible for resistance to vancomycin. In Egypt, relatively lower rates were reported by Sallam et al. [54], who found that 5.9% of *S. aureus* recovered from retail chicken carcasses and chicken products was resistant to vancomycin, while a slightly higher resistance rate of 10.4% was noted by Elshebrawy et al. [53] for *S. aureus* recovered from chicken carcasses, RTE chicken sandwiches, and buffalo milk samples marketed in Egypt. A much higher resistance rate of 80%, however, was reported for vancomycin by *S. aureus* isolated from meat samples tested in Pakistan [59]. Furthermore, Adeyemi et al. [66] found that all *S. aureus* strains (*n* =138) isolated from RTE food samples from the Osogbo metropolis, Southwestern Nigeria, were vancomycin-resistant.

Linezolid and rifampin encountered a complete absence of resistance in our survey, since all of the 560 (100%) *S. aureus* isolates were susceptible to both antibiotics (Table 2). The absolute susceptibility of *S. aureus* isolates to linezolid and rifampin could be attributed to the fact that they were relatively recently approved for treatment by the FDA [67]. A linezolid and rifampin combination can be more effective than either drug alone, especially in infections involving biofilm-producing bacteria. The ability of rifampin to penetrate biofilms and linezolid’s ability to inhibit protein synthesis make them a potent combination. Such a combination can also help in inhibiting the emergence of antibiotic resistance [68]. The absolute susceptibility of *S. aureus* isolates to linezolid and rifampin agrees with a previous study conducted in Korea by Park et al. [69], who confirmed that all *S. aureus* isolates from RTE meals show complete susceptibility to rifampin. However, different resistance rates were reported in other studies worldwide; For instance, Beshiru et al. [65] in Nigeria found that 1.3% and 13.9% of MRSA isolates tested were resistant to linezolid and rifampin, while Chajęcka-Wierzchowska et al. [61] in Poland found that MRSA isolates recovered from various RTE food (cheeses, cured meats, sausages, smoked fishes, salads) had a resistance rate of 6% and 20.5% to linezolid and rifampin, respectively, and Yang et al. [55] in China indicated that 1.6% and 6.5% of MRSA isolated from the best Chinese RTE food were resistant to linezolid and rifampin.

All *S. aureus* strains tested in this study were categorized as multidrug-resistant (MDR) pathogens with an MAR index ranging from 0.2 to 0.7, reinforcing the fact that the antimicrobial resistance phenomenon has become a significant threat to human and veterinary medicine, as it serves as a vehicle for transferring antimicrobial resistance genes to humans, creating serious challenges in patient treatment.

The high MAR index for multidrug-resistant *S. aureus* isolates from RTE pizza indicates a significant public health concern. Food isolates with high MAR indices are more likely to be resistant to standard antibiotic treatments, potentially making infections harder to treat and increasing the risk of severe illness. This also raises concerns about the spread of antibiotic resistance to other bacteria and the potential for resistance genes to transfer to human pathogens. This can lead to prolonged illness, increased healthcare costs, and potentially higher mortality rates.

## 4. Materials and Methods

### 4.1. Collection of Samples

Three hundred ready-to-eat pizza samples (100 each of meat, chicken, and canned tuna pizzas) were collected throughout the period from November 2022 to July 2023 from different restaurants in Mansoura City, Egypt. Each sample was taken in its cardboard box, labeled, and promptly transported to the Food Hygiene, Safety, and Technology Department Laboratory at Mansoura University’s School of Veterinary Medicine in Egypt for microbiological analysis.

### 4.2. Isolation and Identification of Staphylococcus aureus

The isolation and identification of *S. aureus* were carried out according to the International Organization for Standardization (ISO) [70]. Briefly, 25 g from each sample, including the crust, sauce, and topping of the pizza, was aseptically collected and homogenized for 1 min in a sterile homogenizer containing 225 mL of buffered peptone water, from which ten-fold serial dilutions were prepared in sterile test tubes. A volume of 0.2 mL from the appropriate dilutions was spread onto duplicate plates of Baird–Parker selective agar (Oxoid CM275; Oxoid Ltd., Basingstoke, UK) supplemented with egg yolk k-tellurite emulsion, followed by incubation at 35 °C for 48 h. The typical colonies (black, convex, and shiny, surrounded by a halo zone of clearing) were classified as presumptive *S. aureus*. The total count was calculated and recorded as Log_10_ CFU/g of the samples. From each positive sample, 3−5 typical colonies were picked up for further confirmation based on Gram staining, coagulase testing, catalase testing, carbohydrate fermentation, and thermostable nuclease production.

### 4.3. Molecular Analysis

#### 4.3.1. Preparation of Genomic DNA

The genomic DNA was extracted from the confirmed coagulase-positive isolates using a DNA extraction kit (QIAamp DNA Mini Kit; QIAGEN GmbH, Hilden, Germany) according to the manufacturer’s instructions. Genomic DNA from the *E. coli K-12 DH5a* strain was used as a negative control template for PCR analyses.

#### 4.3.2. Molecular Characterization of *S. aureus* Isolates

All 560 *S. aureus* strains isolated from the examined chicken, meat, and tuna pizza samples were screened by PCR for thermostable nuclease (*nuc* gene; the marker gene for *Staphylococcus aureus*), *mec*A (methicillin resistance gene), and vancomycin resistance genes (*van*A and *van*B), as well as selected *S. aureus* virulence genes, *sea*, *seb*, *sec*, *sed*, *hla*, and *tsst*, which encode *Staphylococcus* enterotoxins a, b, c, and d; alpha-hemolysin; and toxic shock syndrome toxin, respectively. The primer sets used for PCR amplification of these genes are listed in Table 4. PCR was performed using the SimpliAmp Thermal Cycler PCR System (Thermo Fisher Scientific Inc.; Waltham, MA, USA). Each 25 µL reaction mixture contained 2 µL *S. aureus* DNA template, 1 µL (6 pmol) each of forward and reverse primers, 12.5 µL ABT 2X Red Mix (Applied Biotechnology; Ismailia, Egypt), and 8.5 µL nuclease-free water. After an initial denaturation at 95 °C for 2 min, 35 amplification cycles consisting of denaturation at 95 °C for 30 s, annealing at 58 °C for 30 s, and extension at 72 °C for 1 min per kbp were performed, followed by a final extension at 72 °C for 5 min. Amplicons were separated by subjecting 7 µL aliquots to agarose (1.2%) gel electrophoresis for 55 min at 95 V, followed by a 20 min staining in ethidium bromide solution. The separated PCR products were then visualized under UV light and photographed.

### 4.4. Antimicrobial Resistance Profile of Staphylococcus aureus Isolates

The antimicrobial resistance profile of the *S. aureus* isolates (*n* = 560) recovered from the different pizza types collected was determined by testing the isolates against 16 antimicrobial agents by agar disc diffusion using Mueller–Hinton agar (Oxoid CM0337) according to the Clinical and Laboratory Standards Institute (CLSI) guidelines, except for oxacillin and vancomycin, which analyzed with the agar dilution method. *S. aureus* strains with vancomycin MIC breakpoints of ≤2, 4–8, and ≥16 μg/mL were verified as susceptible, intermediate, and resistant, respectively [73]. *S. aureus* with oxacillin MIC breakpoints ≤2 µg/mL were considered oxacillin-sensitive, and those with breakpoints ≥4 µg /mL were considered oxacillin-resistant [73].

The sixteen antimicrobials tested (belong to 6 classes) were rifampin (RIF) (30 µg), linezolid (LZ) (30 µg), ciprofloxacin (CIP) (30 µg), gentamycin (GEN) (30 µg), azithromycin (AZM) (30 µg), tetracycline (TE) (30 µg), ampicillin (AMP) (25 µg), penicillin (P) (10 IU), kanamycin (K) (30 µg), cefotaxime (CTX) (30 µg), clindamycin (CD) (2 µg), cefuroxime (CXM) (30 µg), sulfamethoxazole-trimethoprim (25 µg), and nitrofurantoin (F) (300 µg). The inoculated plates covered with the different antimicrobial discs were incubated at 35 °C for 24 h. The susceptibility and resistance categorization were determined based on the diameters of the inhibition zones around the discs according to the National Committee of Clinical Laboratory Standards Institute guidelines [73].

## 5. Conclusions

The current study revealed that about 51% (152/300) of ready-to-eat pizza samples marketed in Mansoura City, Egypt, were contaminated with *S. aureus*. Among the recovered isolates, various enterotoxins (such as *sea*, *seb*, *sec* and *sed*) and the *hla* hemolysin gene were detected at differing frequencies, indicating the potential risk of enterotoxigenic and hemolytic *S. aureus* strains in these products. Additionally, all *S. aureus* isolates (n = 560) recovered from pizza were multidrug-resistant with a mean MAR index of 0.38, demonstrating resistance to penicillin, oxacillin, and ampicillin. Among the isolates, 48.6% (272/560) were identified as MRSA (methicillin-resistant *S. aureus*), and 8.6% (48/560) were identified as VRSA (vancomycin-resistant *S. aureus*), which necessitate critical measures for establishing antimicrobial stewardship programs rationalizing antibiotic usage in veterinary medicine to prevent the spread of such resistant bacteria to humans via the food chain. These findings also represent a great wake-up alarm for public health authorities to set up high hygienic standards in all restaurants and food outlets, including practical hygienic training courses for handlers and routine monitoring of ready-to-eat foods to protect human health in Egypt.

## Figures and Tables

**Figure 1 antibiotics-14-00817-f001:**
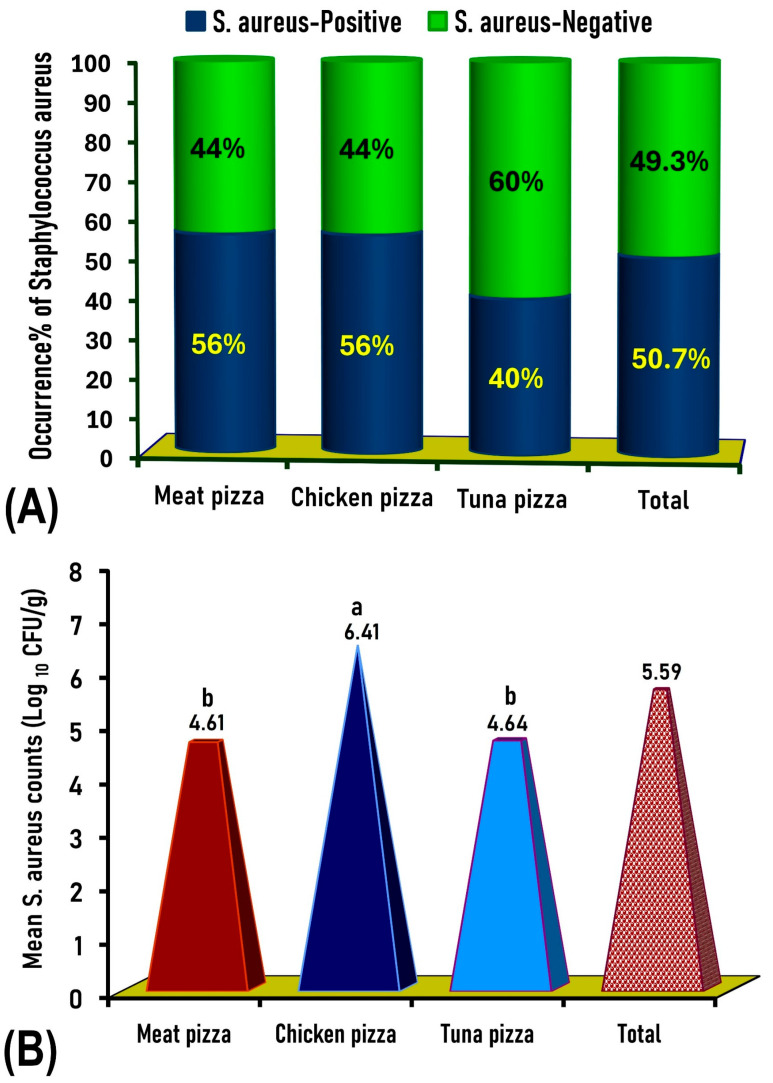
(**A**): Prevalence of *S. aureus* among the examined samples of meat, chicken, and canned tuna pizzas. (**B**): Mean count (log_10_ CFU/g) of *S. aureus* in different types of pizza tested. Mean count values with different letters are significantly different (*p* < 0.05).

**Figure 2 antibiotics-14-00817-f002:**
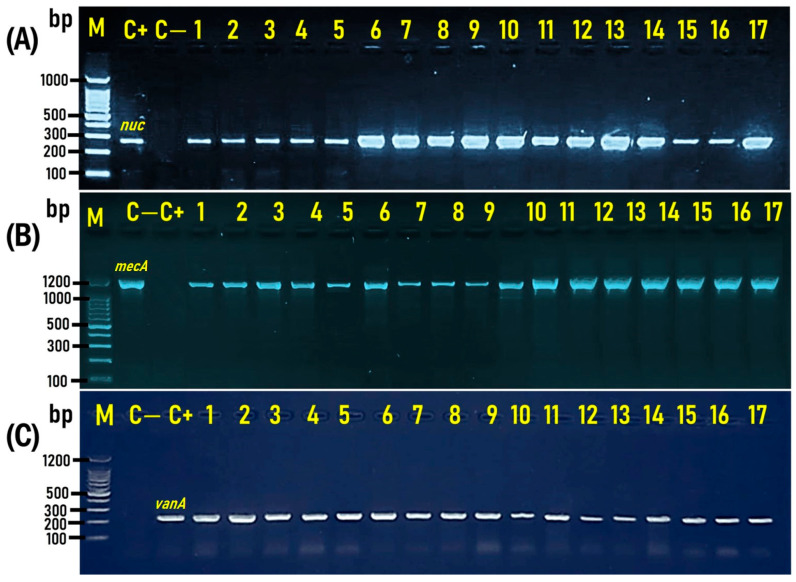
Representative agarose gel electrophoresis for PCR demonstrated amplified bands of the *nuc*, *mecA*, and *vanA* genes. (**A**): Amplified bands of the *nuc* gene-positive *S. aureus* (Lanes 1–17), detected at the expected molecular size of 278 bp. (**B**): The *mecA* gene, which is the specific marker for methicillin-resistant *S. aureus* (MRSA) (Lanes 1 to 17), is detected at an expected molecular size of 1200 bp. (**C**): The *vanA* gene, the specific marker for vancomycin-resistant *S. aureus* (VRSA), is detected at the expected molecular size of 235 bp. M: DNA marker (100 bp gene ladder). C+: Control positive (*S. aureus* ATCC 43300), C—: Control negative (*E*. *coli* K-12 DH5α). Eight microliters of the PCR product were separated by electrophoresis on a 1.5% agarose gel and visualized under UV light.

**Figure 3 antibiotics-14-00817-f003:**
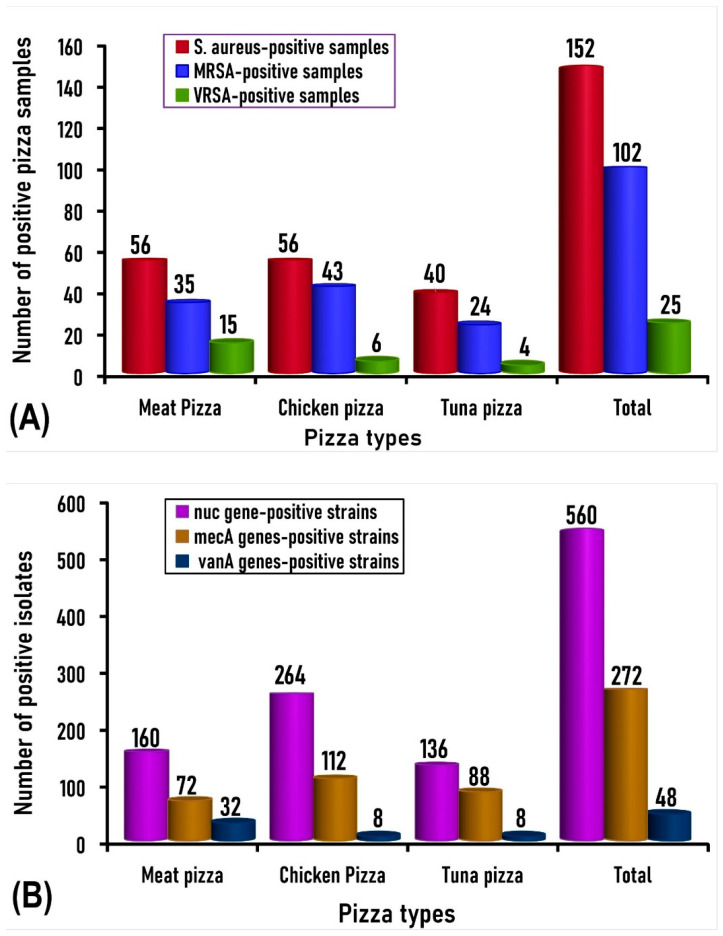
Prevalence and distribution of methicillin-resistant *S. aureus* (MRSA) and vancomycin-resistant *S. aureus* (VRSA) among the nuclease (*nuc*)-positive *S. aureus* isolates (*n* = 560) recovered from the examined samples of meat, chicken, and canned tuna pizza. (**A**): Number of pizza samples positive for *S. aureus*, MRSA, and VRSA. (**B**): Number of isolates identified as *S. aureus*, MRSA, and VRSA based on the existence of *nuc*, *mecA*, and *vanA* genes, respectively.

**Figure 4 antibiotics-14-00817-f004:**
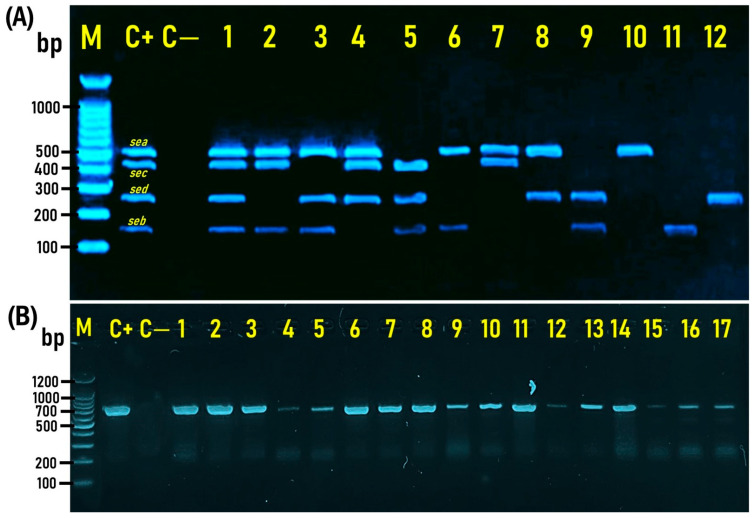
Representative agarose gel electrophoresis for multiplex PCR showing the amplified bands for *S. aureus* enterotoxin genes *sea*, *seb*, *sec*, and *sed*, detected at the expected sizes of 500, 164, 451, and 278 bp, respectively (**A**): Representative isolates positive for *sea*, *seb*, *sec*, and *sed* genes (Lane 1); *sea*, *seb*, and *sec* genes (Lane 2); *sea*, *seb*, and *sed* genes (Lane 3); *sea*, *sec*, and *sed* genes (Lane 4); *seb*, *sec*, and *sed* genes (Lane 5); *sea* and *seb* genes (Lane 6); *sea* and *sec* genes (Lane 7); *sea* and *sed* genes (Lane 8); *seb* and *sed* genes (Lane 9); *sea* gene only (Lane 10); *seb* gene only (Lane 11); and *sed* gene only (Lane 12). PCR-amplified bands of the α-hemolysin (*hla*) gene-positive *S. aureus* (Lanes 1–17) were detected at the expected band size of 704 bp (**B**): M: DNA marker (100 bp gene ladder). C+: Control positive (*S. aureus* ATCC 43300), C—: Control negative (*E*. *coli* K-12 DH5α). The PCR products (8 μL) were separated by electrophoresis on a 1.5% agarose gel and visualized under UV light.

**Figure 5 antibiotics-14-00817-f005:**
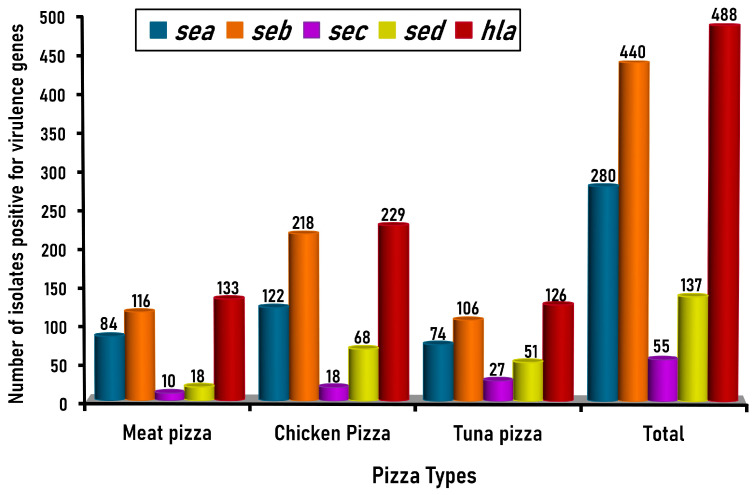
Prevalence and distribution of enterotoxin genes (*sea*, *seb*, *sec*, and *sed*) and the α-hemolysin gene (*hla*) among *S. aureus* isolates (*n* = 560) recovered from the examined samples of meat, chicken, and canned tuna pizza.

**Table 1 antibiotics-14-00817-t001:** Prevalence and count of coagulase-positive *S. aureus* in pizza samples compared with the maximum permissible limit.

Type of Pizza	Number of Samples Tested	Number of Coagulase-Positive *S. aureus* Samples	Coagulase-Positive *S. aureus* Count (Log_10_ CFU/g)		No. (%) of Coagulase-Positive *S. aureus* Exceeding the Maximal Permissible Limits *
			Minimum	Maximum	
Chicken pizza	100	56	2.9	7.7	52 (52%)
Meat pizza	100	56	2.0	5.8	48 (48%)
Canned tuna pizza	100	40	4.2	5.3	40 (40%)
Total	300	152	2.0	7.7	140 (46.7%)

* The maximal permissible limit is 3 log_10_ colony-forming units (CFU)/g for RTE pizza according to Egypt’s National Food Safety Authority [13].

**Table 2 antibiotics-14-00817-t002:** Antimicrobial resistance profiles of *S. aureus* isolates (*n* = 560) recovered from pizza samples.

Antimicrobial	Sensitive	Intermediate	Resistant
Ampicillin (AMP) (25 µg)	0 (0%)	0 (0%)	560 (100%)
Oxacillin (OX) (1 µg)	0 (0%)	0 (0%)	560 (100%)
Penicillin (P) (10IU)	0 (0%)	0 (0%)	560 (100%)
Cefuroxime (CXM) (30 µg)	73 (13%)	0 (0%)	487 (87%)
Cefotaxime (CTX) (30 µg)	129 (23%)	0 (0%)	431 (77%)
Kanamycin (K) (30 µg)	134 (24%)	0 (0%)	426 (76%)
Ciprofloxacin (CIP) (30 µg)	336 (60%)	90 (16%)	134 (24%)
Tetracycline (30 µg)	459 (82%)	23 (4%)	78 (14%)
Azithromycin (AZM) (30 µg)	381 (68%)	134 (24%)	45 (8%)
Clindamycin (CD) (2 µg)	498 (89%)	22 (4%)	40 (7%)
Sulfamethoxazole-trimethoprim (COT) (25 µg)	520 (93%)	0 (0%)	40 (7%)
Gentamycin (GEN) (30 µg)	416 (74%)	120 (22%)	24 (4%)
Nitrofurantoin (F) (300 µg)	375 (67%)	163 (29%)	22 (4%)
Vancomycin (VA) (30 µg)	512 (91.4%)	0 (0%)	48 (8.6%)
Rifampin (RIF) (30 µg)	560 (100%)	0 (0%)	0 (0%)
Linezolid (LZ) (30 µg)	560 (100%)	0 (0%)	0 (0%)

**Table 3 antibiotics-14-00817-t003:** Classification of *S. aureus* isolates (*n* = 560) based on their multiple antibiotic resistance (MAR) index and antimicrobial resistance profiles against the 16 antimicrobial agents tested.

Sources and (Number of Isolates)	Number and (%) of Isolates	Antimicrobial Resistance Pattern	Antimicrobial Classes with Resistance	MAR Index	Classification of Strains
Chicken pizza (6) Meat pizza (6)	12 (2.1%)	AMP-OX-P-CXM-CTX-K-CIP-TE-AZM-CD-COT	Penicillins, cephalosporins, aminoglycosides, fluoroquinolones, tetracyclines, macrolides, lincomycins, sulfonamides	0.688	Multidrug-resistant *Staphylococcus aures*
Meat pizza (34) Canned tuna pizza (10)	44 (7.8%)	AMP-OX-P-CXM-CTX-K-CIP-TE	Penicillins, cephalosporins, aminoglycosides, fluoroquinolones, tetracyclines	0.500	
Chicken pizza (78) Meat pizza (12) Canned tuna pizza (10)	100 (17.9%)	AMP-OX-P-CXM-CTX-K-CIP	Penicillins, cephalosporins, aminoglycosides, fluoroquinolones	0.438	
Chicken pizza (104) Meat pizza (96) Canned tuna pizza (92)	292 (52.1%)	AMP-OX-P-CXM-CTX-K	Penicillins, cephalosporins, aminoglycosides	0.375	
Chicken pizza (22) Meat pizza (12)	34 (6.1%)	AMP-OX-P-CXM-CTX	Penicillins, cephalosporins	0.313	
Chicken pizza (32) Canned tuna pizza (24)	56 (10%)	AMP-OX-P-CXM	Penicillins, cephalosporins	0.250	
Chicken pizza (22)	22 (4%)	AMP-OX-P	Penicillins	0.188	
Sum = 560	Mean MAR Index for all isolates = 0.375				

**Table 4 antibiotics-14-00817-t004:** Primer set sequences used for molecular characterization of *Staphylococcus aureus* isolates recovered from pizza.

Target Gene	Primer Direction and Sequence	Amplicon Size (bp)	Reference
*nuc*	F: 5′-gcgattgatggtgatacggtt-3′ R: 5′-agccaagccttgacgaactaa-3′	278	This study
*hla*	F: 5′-gaagtctggtgaaaaccctga-3 R: 5′-tgaatcctgtcgctaatgcc-3′	704	Xiaohong and Yanjun [71]
*sea*	F: 5′-tgcagggaacagctttaggcaa-3′ R: 5′-gattaatcccctctgaaccttcc-3′	500	Sallam et al. [54]
*seb*	F: 5′-gtatggtggtgtaactgagc-3′ R: 5′-ccaaatagtgacgagttagg-3′	164	Mehrotra et al. [72]
*sec*	F: 5′-agatgaagtagttgatgtgtatgg-3′ R: 5′-cacacttttagaatcaaccg-3′	451	Mehrotra et al. [72]
*sed*	F: 5′-ccaataataggagaaaataaaag-3′ R: 5′-attggtattttttttcgttc-3′	278	Mehrotra et al. [72]
*tsst-1*	F: 5′-ctagactggtatagtagtggg-3′ R: 5′-cgccacttatttggaaatgg-3′	235	This study
*mecA*	F: 5′-gattgggatcatagcgtca-3′ R: 5′-cagtatttcaccttgtccg-3′	1200	Sallam et al. [54]
*vanA*	F: 5′-gggaaaacgacaattgc-3′ R: 5′-gtacaatgcggccgtta-3′	235	This study

## Data Availability

All data supporting the findings of this study is included within the article. Any additional information is available from the corresponding author upon reasonable request.

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
