# Peer review of "Prevalence and Characterization of the Antimicrobial Resistance and Virulence Profiles of Staphylococcus aureus in Ready-to-Eat (Meat, Chicken, and Tuna) Pizzas in Mansoura City, Egypt"

_antibiotics, 2025, doi:10.3390/antibiotics14080817_

Round 1

Reviewer 1 Report

Comments and Suggestions for Authors

In this paper, the authors investigate the contamination rate of various categories of ready-to-eat pizzas and especially focus on characterizing the resistant profiles of S. aureus in Mansoura, Egypt.

The paper is comprehensive however my biggest feedback is how the text has ben structured. Having a separate, detailed "discussion" section would significantly improve the readability of the manuscript. Right now, the results are somewhere buried in the thick discussion and hence separating results from discussion would be a better structure to clearly present the findings.

Line 93-94: Needs appropriate reference 

Itlaicize "S. aureus" in line 190, 191, 284

Just like Fig. 4a, it would be helpful to have the gene name over the corresponding bands in Figure 2a, 2b, 2c.

Author Response

Reviewer#1

In this paper, the authors investigate the contamination rate of various categories of ready-to-eat pizzas and especially focus on characterizing the resistant profiles of S. aureus in Mansoura, Egypt.

Comment 1: The paper is comprehensive however my biggest feedback is how the text has been structured. Having a separate, detailed "discussion" section would significantly improve the readability of the manuscript. Right now, the results are somewhere buried in the thick discussion and hence separating results from discussion would be a better structure to clearly present the findings.

Response: We sincerely thank the reviewer for the valuable suggestion regarding the structure of the manuscript. In response to this comment, we have revised the manuscript to clearly separate the Results and Discussion sections. This restructuring improves the clarity and flow of the manuscript by distinctly presenting the findings before discussing their implications. We believe this change enhances the overall readability and allows for a more focused interpretation of the results, as recommended.

Comment 2: Line 93-94: Needs appropriate reference.

Response: Thank you for pointing this out. An appropriate reference has been added to support the statement in lines 93–94. The following citation has been included:
Scallan, E., Hoekstra, R. M., Angulo, F. J., Tauxe, R. V., Widdowson, M.-A., Roy, S. L., Jones, J. L., & Griffin, P. M. (2011). Foodborne illness acquired in the United States—major pathogens. Emerging Infectious Diseases, 17(1), 7–15.

Comment 3: Italicize "S. aureus" in line 190, 191, 284.

Response: Thank you for your careful observation. We have italicized S. aureus as requested in lines 190, 191, and 284 to ensure correct scientific formatting.

Comment 4: Just like Fig. 4a, it would be helpful to have the gene name over the corresponding bands in Figure 2a, 2b, 2c.

Response: Thank you for the helpful suggestion. Gene names have now been added above the corresponding positive control bands in Figures 2a, 2b, and 2c to enhance clarity and consistency with Figure 4a.

Reviewer 2 Report

Comments and Suggestions for Authors

Elsalkh et al. have aimed to determine the prevalence of S. aureus, virulence genes, and antimicrobial resistance profiles in ready-to-eat pizzas marketed in Mansoura, Egypt. This study contains some important data and interpretations useful for the readers in the field. However, the following issues need to be addressed for a possible publication in Antibiotics.

  1. The title should be modified as “Prevalence and Characterization of Methicillin- and Vancomycin-resistant Staphylococcus aureus in Meat, Chicken, and Tuna Pizzas”
  2. In the abstract, the potential public health implications should be mentioned clearly. Also mention the sample size much earlier in the abstract.
  3. Line 56-64: description on pizza ingredients and preparation should be condensed to be concise.
  4. Line 81-90: instead of providing general microbiological details, this paragraph should focus on describing S. aureus as a foodborne pathogen.
  5. Line 124-128: clarify the novelty on focusing on pizzas by highlighting the scarcity of data on S. aureus contamination in pizzas.
  6. Line 158-184: Although the comparison with previous studies have been extensive, they lack critical analysis on why the variations occur. Especially, why there is a relatively high contamination rates in Egypt in spite of general hygiene assumptions.
  7. Line 194-196: the description of molecular marker genes should be simplified to focus on the study findings.
  8. Figures 2 and 3 should have better captions to explain the key implications and not just procedural details.
  9. Line 249-253: the more detailed molecular mechanisms (especially functions of mecA and vanA genes) should be condensed to streamline the focus in relevance to this study.
  10. Explain the public health implications of 48.6% MRSA and 8.6% VRSA more clearly and discuss briefly how this prevalence compares internationally.
  11. Line 299-314: the surplus details of PCR band patterns especially procedural details should be move to the supplemental material.
  12. Line 329-332: the public health implications of enterotoxins (seb and sea genes) should be emphasized.
  13. Line 417-440: Although the comparison with previous studies is useful, they should be condensed to be more concise.
  14. Line 496-504: the public health implications of high MAR index should be strongly emphasized.
  15. Please discuss why linezolid and rifampin remain effective.
  16. Table 4 containing the primer sequences should be moved to the supplemental material.
  17. Lines 527-541: the description of isolation and identification should be streamlined to remove basic microbiology steps unless deviations from standard protocol exist.
  18. Line 592-596: Important actionable recommendations and control measures should be provided by proposing specific strategies including improved hygienic training, routing monitoring of ready-to-eat foods and antimicrobial stewardship programs.
  19. Figures 1-5: the captions should be improved for clarity and more interpretive.

Author Response

Reviewer#2

Elsalkh et al. have aimed to determine the prevalence of S. aureus, virulence genes, and antimicrobial resistance profiles in ready-to-eat pizzas marketed in Mansoura, Egypt. This study contains some important data and interpretations useful for the readers in the field. However, the following issues need to be addressed for a possible publication in Antibiotics.

  1. The title should be modified as “Prevalence and Characterization of Methicillin- and Vancomycin-resistant Staphylococcus aureus in Meat, Chicken, and Tuna Pizzas”.

Response: Thank you for the suggestion. The title has been modified accordingly to: “Prevalence and Characterization of Methicillin- and Vancomycin-resistant Staphylococcus aureus in Meat, Chicken, and Tuna Pizzas.”

  1. In the abstract, the potential public health implications should be mentioned clearly. Response: Such issue is covered and added in the abstract of the revised manuscript (Lines 16-20).

Also mention the sample size much earlier in the abstract.

Response: The sample size is identified early in the revised manuscript according to your suggestion.

  1. Line 56-64: description on pizza ingredients and preparation should be condensed to be concise.

Response: Thank you for the suggestion. The description of pizza ingredients and preparation in lines 56–64 has been revised to be more concise in the updated manuscript.

  1. Line 81-90: instead of providing general microbiological details, this paragraph should focus on describing S. aureus as a foodborne pathogen.

Response: Thank you for your valuable feedback. This paragraph (lines 81–90) has been edited in the revised manuscript to focus specifically on Staphylococcus aureus as a foodborne pathogen, rather than providing general microbiological details.

  1. Line 124-128: clarify the novelty on focusing on pizzas by highlighting the scarcity of data on S. aureus contamination in pizzas.

Response: Thank you for the insightful comment. The revised manuscript now highlights the novelty of focusing on pizzas by emphasizing the scarcity of available data on Staphylococcus aureus contamination in ready-to-eat pizzas, as suggested.

  1. Line 158-184: Although the comparison with previous studies have been extensive, they lack critical analysis on why the variations occur. Especially, why there is a relatively high contamination rates in Egypt in spite of general hygiene assumptions?

Response: The explanation of the higher prevalence rate of S. aureus in RTE pizza contamination in the present study refers to the possible post-cooking contamination from the pizza handlers, which may be left aside for some time at room temperature before being packaged and delivered to consumers, and the difference in cultural, educational, and sanitary levels of food handlers. Such an explanation is already written (Lines 158-162), and Lines 186-190.

  1. Line 194-196: the description of molecular marker genes should be simplified to focus on the study findings.

Response: The description of molecular marker genes is simplified in the revised manuscript.

  1. Figures 2 and 3 should have better captions to explain the key implications and not just procedural details.

Response: The figure captions are improved in the revised manuscript.

  1. Line 249-253: the more detailed molecular mechanisms (especially functions of mecA and vanA genes) should be condensed to streamline the focus in relevance to this study. Response: Thank you for the suggestion. The detailed discussion of molecular mechanisms in lines 249–253 has been condensed in the revised manuscript to streamline the focus for this study. We retained a brief, targeted description of the functions of mecA and vanA only as they directly relate to methicillin and vancomycin resistance, respectively, and removed extraneous mechanistic detail.
  2. Explain the public health implications of 48.6% MRSA and 8.6% VRSA more clearly and discuss briefly how this prevalence compares internationally.

Response: A paragraph concerning the public health implications of 48.6% MRSA and 8.6% VRSA is added in the revised manuscript (Lines 230-234). The comparison of the current prevalence rates of MRSA and VRSA with other rates reported internationally in other publications is already stated in the discussion section (Lines 246-254) and Lines (276-280).

  1. Line 299-314: the surplus details of PCR band patterns especially procedural details should be move to the supplemental material.

Response: The legend has been revised and simplified to 10 lines from 15 lines.

  1. Line 329-332: the public health implications of enterotoxins (seb and sea genes) should be emphasized.

Response: The public health implications of seb and sea genes is added in the revised manuscript (Lines 342-350).

  1. Line 417-440: Although the comparison with previous studies is useful, they should be condensed to be more concise.

Response: Such a paragraph includes 5 references discussing the comparison of 3 classes of antimicrobials.

  1. Line 496-504: the public health implications of high MAR index should be strongly emphasized.

Response: The public health implications of the high MAR index are emphasized in the revised manuscript (Lines 520-526).

  1. Please discuss why linezolid and rifampin remain effective.

Response: Discussion concerning the susceptibility of S aureus toward such 2 antibiotics is written (Lines: 502-508).

  1. Table 1 containing the primer sequences should be moved to the supplemental material. Response: Thank you for the suggestion. We understand the reasoning behind moving Table 1 to the supplementary material. However, we prefer to retain it in the main text, as it includes primer sequences along with four important references that are directly relevant to the methodology. Keeping the table in the main manuscript ensures easier accessibility for readers and supports transparency and reproducibility of the experimental design.
  2. Lines 527-541: the description of isolation and identification should be streamlined to remove basic microbiology steps unless deviations from standard protocol exist. Response: The isolation method is condensed to 11 lines in the revised manuscript instead of 17 lines.
  3. Line 592-596: Important actionable recommendations and control measures should be provided by proposing specific strategies, including improved hygienic training, routine monitoring of ready-to-eat foods, and antimicrobial stewardship programs.

Response: The recommendations in the conclusion section have been improved based on your valuable suggestions.

  1. Figures 1-5: the captions should be improved for clarity and more interpretive.

Response: Figure captions have been simplified and improved.

Reviewer 3 Report

Comments and Suggestions for Authors

The present work „Methicillin- and vancomycin-resistant Staphylococcus aureus in meat, chicken, and tuna pizzas” written by Sara Amgad Elsalkh, Amira Ibrahim Zakaria, Samir Mohammed Abd-Elghany, Kálmán Imre, Adriana Morar, Khalid Ibrahim Sallam, is related to the  investigation of  meat, chicken, and tuna pizzas marketed in Egypt, followed by the characterization of the virulence genes (sea, seb, sec, sed, and tsst-1) and the antimicrobial resistance profile of the strains isolated during this survey.

Staphylococcus aureus is a species world known to be the cause of many food born diseases due to the smart pathogenic factors and the various enterotoxins that it possesses. Nowadays, fast food is linked to a wide variety of serious health problems globally. The research team emphasizes on the quality of the used raw materials in the fast foods, unclean utensils, poor personal hygiene of workers, insufficient cooking temperature, and post-processing contamination.

The problem in the last decades stems from the extensive misuse of antibiotics in food-producing animals, in therapeutic activities and productive purposes. This has led to the development of antimicrobial resistance genes that are entering the human food supply chain. World Health Organization lists S. aureus as a high-priority bacterium.

The present investigation is important to be published because S. aureus has developed multiple resistance genes to overcome the mostly used antibiotics in the treatment of infections. The highlight of the authors here is that the state rulers must take into consideration all of the scientific reports in regard to the food processing, and emphasize the need for stricter antibiotic regulations and the development of new antimicrobial agents that would treat most microbial intoxications successfully.

Here is my opinion:

The manuscript’s topic is relevant to the current problems of the industrial fast food production and the linked health problems due to the treatment difficulties of S. aureus diseases. The authors managed to establish the antimicrobial resistance profile of S. aureus isolated from different types of ready-to-eat pizza. It is worth being published in Antibiotics.

The list of literature is presented by quite current studies – almost 60% of them are from the last 10 years, and almost 30% are from the last 5 years. This shows relevance of the research done by the authors with the most current investigations.

The manuscript is of need for the following corrections:

  • In the PDF file my suggestions are colored in orange.
  • In my opinion, the English language is quite good and needs just a few small corrections.

As a reviewer, I state that I do not have conflicts of interest with the authors of this research work.

Author Response

Reviewer#3

The present work „Methicillin- and vancomycin-resistant Staphylococcus aureus in meat, chicken, and tuna pizzas” written by Sara Amgad Elsalkh, Amira Ibrahim Zakaria, Samir Mohammed Abd-Elghany, Kálmán Imre, Adriana Morar, Khalid Ibrahim Sallam, is related to the  investigation of  meat, chicken, and tuna pizzas marketed in Egypt, followed by the characterization of the virulence genes (sea, seb, sec, sed, and tsst-1) and the antimicrobial resistance profile of the strains isolated during this survey.

Staphylococcus aureus is a species world known to be the cause of many food born diseases due to the smart pathogenic factors and the various enterotoxins that it possesses. Nowadays, fast food is linked to a wide variety of serious health problems globally. The research team emphasizes on the quality of the used raw materials in the fast foods, unclean utensils, poor personal hygiene of workers, insufficient cooking temperature, and post-processing contamination.

The problem in the last decades stems from the extensive misuse of antibiotics in food-producing animals, in therapeutic activities and productive purposes. This has led to the development of antimicrobial resistance genes that are entering the human food supply chain. World Health Organization lists S. aureus as a high-priority bacterium.

The present investigation is important to be published because S. aureus has developed multiple resistance genes to overcome the mostly used antibiotics in the treatment of infections. The highlight of the authors here is that the state rulers must take into consideration all of the scientific reports in regard to the food processing, and emphasize the need for stricter antibiotic regulations and the development of new antimicrobial agents that would treat most microbial intoxications successfully.

Here is my opinion:

The manuscript’s topic is relevant to the current problems of the industrial fast food production and the linked health problems due to the treatment difficulties of S. aureus diseases. The authors managed to establish the antimicrobial resistance profile of S. aureus isolated from different types of ready-to-eat pizza. It is worth being published in Antibiotics.

The list of literature is presented by quite current studies – almost 60% of them are from the last 10 years, and almost 30% are from the last 5 years. This shows relevance of the research done by the authors with the most current investigations.

The manuscript is of need for the following corrections:

  • In the PDF file my suggestions are colored in orange.
  • In my opinion, the English language is quite good and needs just a few small corrections.

As a reviewer, I state that I do not have conflicts of interest with the authors of this research work.

Response: We sincerely thank the reviewer for the thorough evaluation and positive assessment of our manuscript titled “Methicillin- and vancomycin-resistant Staphylococcus aureus in meat, chicken, and tuna pizzas.” We appreciate your recognition of the relevance and importance of our work, as well as your encouraging remarks regarding the literature and overall quality.

Regarding your specific suggestions and the corrections marked in the attached PDF file (highlighted in orange), we have carefully reviewed and addressed each point. All requested corrections have been successfully incorporated into the revised manuscript. We have ensured that the language improvements enhance clarity and readability while preserving the scientific accuracy and flow of the text.

We are grateful for your constructive feedback, which has undoubtedly improved the quality of our paper. We look forward to your continued guidance and hope that the revised version now meets the journal’s standards for publication.

Round 2

Reviewer 2 Report

Comments and Suggestions for Authors

The authors have satisfactorily addressed all the comments raised by reviewers and substantially improved the overall quality of the article. Therefore, I recommend accepting this article for publication in Antibiotics.

Author Response

Thank you for your time and review!